# Evolution of sequence-specific anti-silencing systems in *Arabidopsis*

Aoi Hosaka[1,2], Raku Saito[1,2], Kazuya Takashima[1], Taku Sasaki[1,3], Yu Fu[1,2], Akira Kawabe [4], Tasuku Ito[1,3], Atsushi Toyoda [5], Asao Fujiyama[5], Yoshiaki Tarutani[1,2] & Tetsuji Kakutani[1,2,3]

The arms race between parasitic sequences and their hosts is a major driving force for evolution of gene control systems. Since transposable elements (TEs) are potentially deleterious, eukaryotes silence them by epigenetic mechanisms such as DNA methylation. Little is known about how TEs counteract silencing to propagate during evolution. Here, we report behavior of sequence-specific anti-silencing proteins used by *Arabidopsis* TEs and evolution of those proteins and their target sequences. We show that VANC, a TE-encoded anti-silencing protein, induces extensive DNA methylation loss throughout TEs. Related VANC proteins have evolved to hypomethylate TEs of completely different spectra. Targets for VANC proteins often form tandem repeats, which vary considerably between related TEs. We propose that evolution of VANC proteins and their targets allow propagation of TEs while causing minimal host damage. Our findings provide insight into the evolutionary dynamics of these apparently "selfish" sequences. They also provide potential tools to edit epigenomes in a sequence-specific manner.

[1] Department of Integrated Genetics, National Institute of Genetics, Yata 1111, Shizuoka 411-8540, Japan. [2] Department of Genetics, School of Life Science, SOKENDAI (The Graduate University for Advanced Studies), Yata 1111, Shizuoka 411-8540, Japan. [3] Department of Biological Sciences, Graduate School of Science, The University of Tokyo, Hongo, Bunkyo-ku, Tokyo 113-0033, Japan. [4] Department of Bioresource and Environmental Sciences, Faculty of Life Sciences, Kyoto Sangyo University, Motoyama Kamigamo, Kyoto 606-8555, Japan. [5] Center for Information Biology, National Institute of Genetics, Yata 1111, Shizuoka 411-8540, Japan. Aoi Hosaka and Raku Saito contributed equally to this work. Correspondence and requests for materials should be addressed to A.H. (email: ahosaka@nig.ac.jp) or to T.K. (email: tkakutan@nig.ac.jp)

The arms race between parasitic sequences and hosts is a major driving force for evolution of gene control systems. As a defense against parasitic sequences such as viruses and transposable elements (TEs), hosts employ mechanisms such as RNAi, chromatin modifications, and DNA methylation[1–11]. Viruses in turn often deploy anti-defense mechanisms[12–15]. Anti-defense mechanisms are widespread in pathogens but the target specificity of these mechanisms is generally low and non-specific anti-defense strategies often reduce host fitness severely. Non-specific anti-defense is less common in TEs, likely reflecting their life cycle in which they generally remain in the same host and depend on host survival. Although horizontal transfer of TEs is known[16,17], this apparently occurs rarely compared to horizontal transfer of viruses, and it is generally thought to be important for TEs to proliferate while avoiding damage to their host. Despite a major impact of TEs on genome evolution, little is known about strategies of TEs to counteract silencing and propagate.

*Arabidopsis* serves as an ideal model organism to investigate control of TEs, with precise TE sequences throughout the genome and *trans*-acting mutations affecting TE activity[6,8]. We have previously reported that an *Arabidopsis* TE, named *Hiun* (*Hi*), is normally silenced by DNA methylation but has an activity to counteract this silencing[9,18]. Expression of VANC, one of the proteins encoded in *Hi*, induces transcriptional derepression of *Hi*-encoded genes and mobilization of *Hi*. In addition, when full-length *Hi* is transformed into wild-type *Arabidopsis* plants, the transgene induces loss of DNA methylation in the entire *Hi*. DNA methylation in other TEs is unaffected. A very enigmatic feature of this demethylation activity is how the specificity can be determined for such long target sequences.

Another intriguing feature of anti-silencing is its evolution. *Hi* belongs to a TE family called *VANDAL21*, which is one of multiple *VANDAL* TE families found in the *Arabidopsis* genome[18–20]. Although *Hi* transgenes induce DNA methylation loss in *VANDAL21* copies, other *VANDAL* copies are unaffected. Still, many other *VANDAL* family members encode proteins related to VANC.

Here, we report behavior and evolution of the anti-silencing proteins VANC and their target sequences. After showing that VANC is sufficient for inducing sequence-specific loss of DNA methylation in the entire length of target TEs, we show that VANC proteins have evolved to induce loss of DNA methylation in TEs of completely different spectra. The most enigmatic features of VANC function are how one protein can recognize the long targets in a very sequence-specific manner and yet could have evolved to change targets. We show that tandem repeat formation is central to the specificity of related VANC proteins. VANC binds a short DNA motif in vivo and in vitro. This motif appears to have accumulated and evolved together by tandem repeat formation. We propose that through coevolution of VANC proteins and target DNA motifs, these *VANDAL* TEs escaped epigenetic silencing to propagate while causing minimal host damage.

## Results

**VANC protein evolves to affect different target TEs.** We have previously reported that a full-length transgene of *Hi* induces DNA methylation loss at endogenous *VANDAL21* copies[18]. Strikingly, entire TEs of several kilobases in length are extensively hypomethylated in the *Hi* transgenic lines[18]. This raises the question how such long sequences can be hypomethylated in a sequence-specific manner. One possibility is that the sequence specificity is defined directly by the nucleotide sequence of the transgene, rather than by the encoded protein. In order to see if regions other than *VANC* are involved in sequence specificity, we examined the hypomethylation effect of a transgene that lacked the other open reading frames (ORFs) of *Hi* (*VANA21* and *VANB21*) (ΔAB; Fig. 1a). Whole-genome bisulfite sequencing (WGBS) revealed that ΔAB induced sequence-specific loss of methylation very similar to that of the full-length *Hi* (Fig. 1b–d; Supplementary Fig. 1; Supplementary Data 1), suggesting that *VANC* expression is sufficient for inducing a loss of DNA methylation at the endogenous *VANDAL21* elements. In order to test if the VANC protein, rather than the transcribed RNA, is responsible for the induced anti-silencing effect, we examined the effect of a *VANC* transgene with nonsense mutations within the coding sequences (Fig. 1a). For both of two transgenes with nonsense mutations in the *VANC* ORF, the anti-silencing effects were abolished; they did not induce transcriptional derepression or mobilization of the endogenous *Hi* (Supplementary Fig. 2). These observations suggest that VANC (hereafter referred to as VANC21) protein function is responsible for the loss of DNA methylation.

Although VANC21 induced hypomethylation in *VANDAL21* members specifically, proteins related to VANC21 are encoded in many other *VANDAL* family members[18]. In order to detect the effects of these VANC21-related proteins, we introduced one of them, a *VANC21*-like gene in *VANDAL6* (AT4G09370, hereafter referred to as *VANC6*), into wild-type plants by transformation (Fig. 1a, bottom). WGBS of the *VANC6* transgenic lines revealed that *VANDAL6* copies and related *VANDAL* family members, such as *VANDAL8* copies, were hypomethylated in them (Fig. 1e, f; Supplementary Fig. 3). Again, the hypomethylation was sequence-specific. Importantly, the spectra of hypomethylated TEs were completely different between *VANC21* and *VANC6* transgenic lines (Fig. 1g). In addition to the hypomethylation of *VANDAL6* and related copies, VANC6 induced transcriptional derepression in these hypomethylated loci (Supplementary Fig. 4). These results show that these *VANC* genes have evolved to induce anti-silencing that is specific for the TE sequences similar to that of the copy encoding the VANC. It is mysterious how the sequence specificity is defined for such long targets that share high sequence similarity. The separation of the *VANDAL* families during evolution was relatively recent[18], and therefore the anti-silencing mechanism must differentiate efficiently among closely related sequences.

**VANC21 is localized in non-coding regions in *VANDAL21*.** To examine how VANC proteins function, we determined VANC21 localization within the genome using chromatin immunoprecipitation followed by sequencing (ChIP-seq). Chromatin from transgenic plants expressing FLAG-tagged *VANC21* was immunoprecipitated with anti-FLAG antibody and associated DNA was sequenced. The ChIP signal was highly enriched at *VANDAL21* loci (Fig. 2a; Supplementary Fig. 5). *VANDAL21*-specific localization was also confirmed by ChIP-seq using an antibody against the intact VANC21 protein (Supplementary Fig. 5d). Interestingly, the accumulation of VANC21 was not uniform; stronger signals were found in non-coding regions within *VANDAL21* copies (Fig. 2b; Supplementary Fig. 6), such as introns, intergenic regions, and terminal non-coding regions.

VANC21 localization matched well with its effect on DNA methylation. The hypomethylation effect of VANC21 is generally stronger in non-CG sites compared to CG sites (Figs. 1b, 2b). Although the entire *VANDAL21* sequence tends to lose DNA methylation in non-CG sites, the hypomethylation effect on CG sites tends to be local (Figs. 1b, 2b). This local hypomethylation effect corresponds closely to the localization of VANC21 protein (Fig. 2b, c). The localization also corresponds to hypomethylation of non-CG sites, but the effect is broader than that observed at

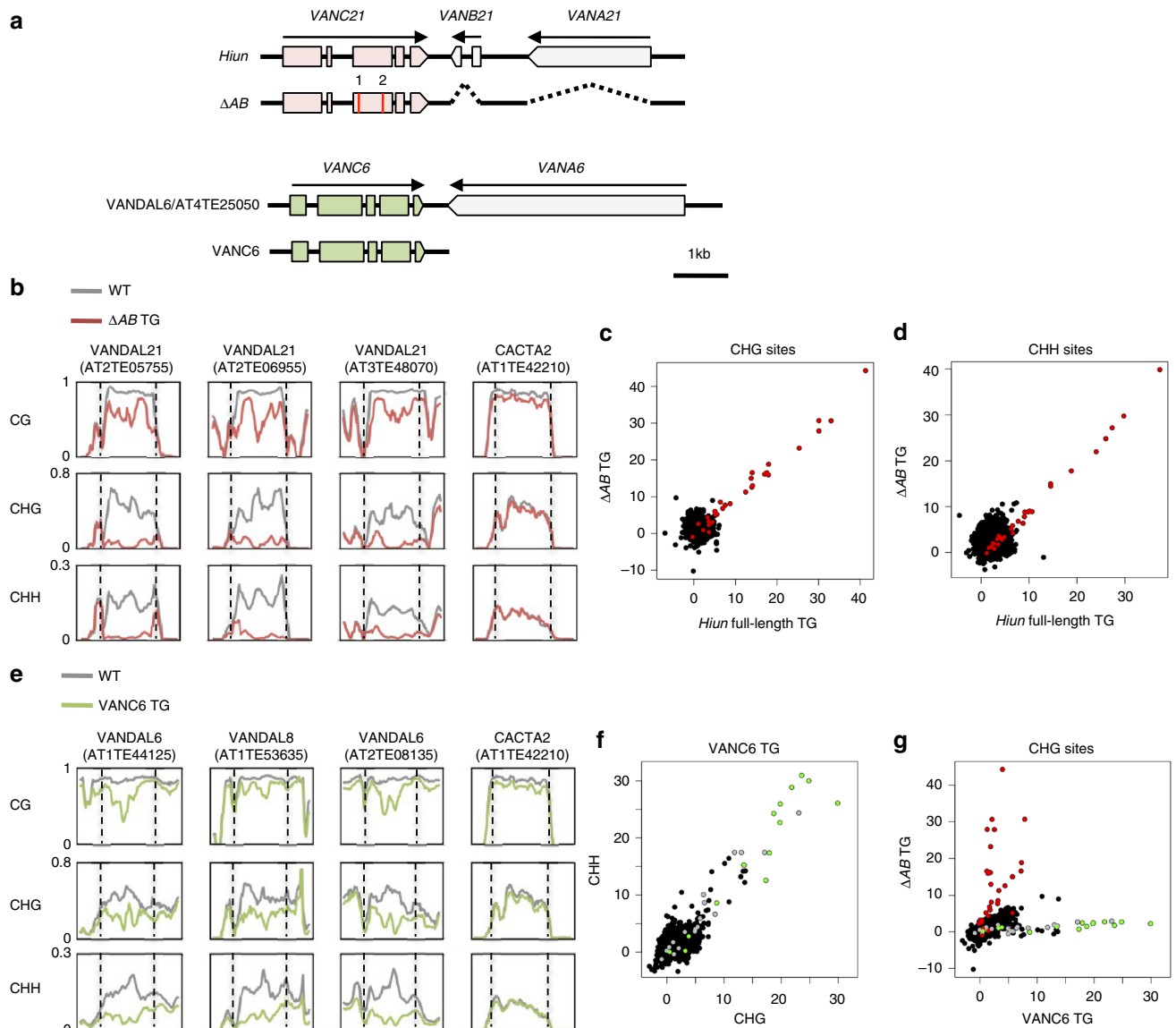

**Fig. 1** Sequence-specific hypomethylation by VANC proteins. **a** Schematic diagram of structures of *VANDAL* transposons and the modified transgenes used. Boxes indicate exons. Vertical red lines show the positions of two nonsense mutations in *VANC21* (Supplementary Fig. 2). **b** DNA methylation levels of *VANDAL21* copies in Δ*AB* transgenic plants and parental wild-type plants (WT). Broken lines show TE ends. Each point represents proportion of methylated cytosine for a sliding window with seven fractions after separating each TE for 100 fractions. Right and left flanking regions are also analyzed by the same conditions. **c**, **d** Comparison of DNA hypomethylation between full-length *Hi* and Δ*AB* transgenic plants for CHG (**c**) and CHH sites (**d**). *VANDAL21* copies are colored red. **e**, **f** Hypomethylation effects of *VANC6* transgene. In **e**, conditions are as in **b**. In **f**, DNA hypomethylation is shown for each TE at CHG sites and CHH sites. *VANDAL6* copies and *VANDAL8* copies are colored green and gray, respectively. **g** Comparison of DNA hypomethylation between Δ*AB* and *VANC6* transgenic plants at CHG sites. Results at CHH sites are shown in Supplementary Fig. 1f–h. In the panels **c**, **d**, **f**, and **g**, TEs more than 1 kb long are plotted (*N* = 5866). The significance of decrease in DNA methylation was assessed by the value $(Mn/Cn - Mt/Ct)/(1/\sqrt{Cn} + 1/\sqrt{Ct})$, where $Mn$, $Cn$, $Mt$, and $Ct$ are methylated cytosine ($M$) and total cytosine ($C$) counts mapped for each TE in the non-transgenic ($n$) and transgenic ($t$) plants, respectively[18]. This value shows the significance by weighing the change in the methylation ratio with root of the count number. Effects of Δ*AB* and *VANC6* transgenes on DNA methylation status of TEs longer than 1 kb are also shown in Supplementary Data 1

CG sites (Fig. 2d, e), reflecting a spread of hypomethylation to surrounding regions extending to entire TEs (Figs. 1b, 2b).

**VANC21 is bound to specific motifs in vivo and in vitro**. To understand how VANC21 determines its targets, we searched for sequences statistically overrepresented in regions where VANC21 localized. A nine-base motif, "YAGTATTAY (Y = T or C)" was the most overrepresented candidate motif (Fig. 3a; Supplementary Table 1). Next, we examined the ability of VANC21 to bind DNA containing this motif in vitro by electrophoretic mobility

shift assay (EMSA). Consistent with the prediction from ChIP-seq, VANC21 induced a mobility shift for double-stranded DNA with sequences of five non-coding regions with this motif within *VANDAL21* (probes 1, 2, 3, 6, and 7 in Fig. 3b, c, and Supplementary Fig. 7). In all of these 40-bp probes, single-base substitutions within the YAGTATTAY motif resulted in drastic reductions of the protein-binding efficiency, suggesting that this motif is indeed important for efficient binding of VANC21. The binding was much less efficient for sequences in coding regions within *VANDAL21* (probes 4 and 5 in Fig. 3b, c), which do not have the YAGTATTAY motif. Taken together, these results

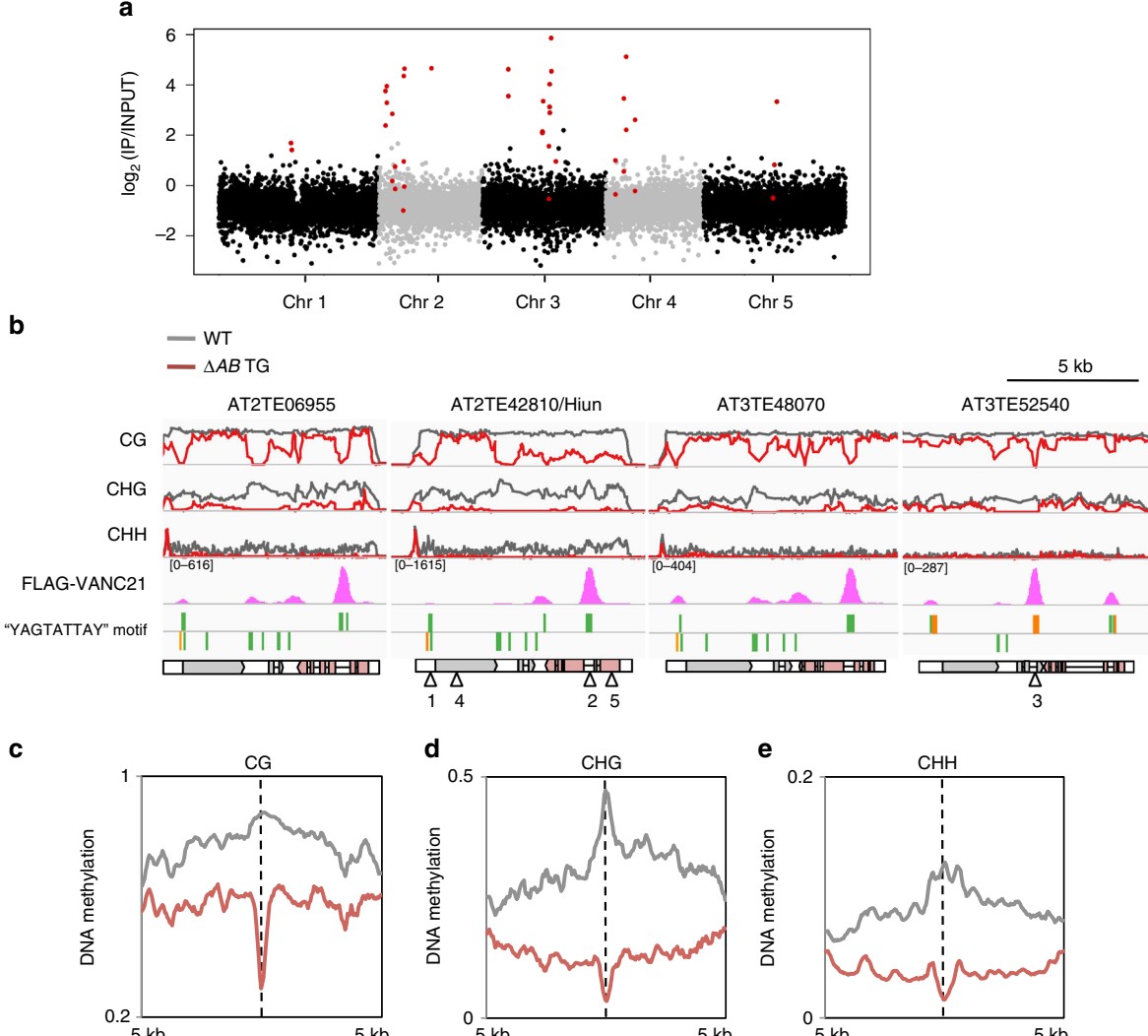

**Fig. 2** Genomic localization of the VANC21 protein. **a** A genome-wide view showing the enrichment of FLAG-VANC21 signal. Each dot represents signal in a 10 kb region. Red dots indicate the regions with *VANDAL21* copies more than 1 kb long. **b** Genome browser views showing the FLAG-VANC21 signals with normalized coverages (per million mapped reads) and DNA methylation profiles (0–100%) of WT and Δ*AB* transgenic plants at *VANDAL21* copies. Each point represents proportion of methylated cytosine counted within five successive cytosine residues. *VANC21* exons are colored red. Arrowheads indicate probe sites used in EMSA (Fig. 3). **c–e** Three contexts of DNA methylation level around VANC21 biding loci. Around the summits of VANC21-binding loci within *VANDAL21* ($N = 89$), 500 bp-binned averages of DNA methylation profiles were plotted for WT and Δ*AB* transgenic plants with steps of 50 bp

suggest that VANC21 was directed to these non-coding regions of *VANDAL21* copies through recognition of specific DNA sequences.

Within the *Arabidopsis thaliana* genome, this motif, YAG-TATTAY, was highly enriched in *VANDAL21* copies, but it is rarely found in other TEs, including other *VANDAL* family members (Fig. 3d; Supplementary Fig. 8a). The motif was mainly found in non-coding regions of *VANDAL21* copies, which is consistent with the VANC21-binding pattern (Fig. 2b; Supplementary Fig. 6). As the YAGTATTAY sequence is short, this motif is also found sporadically outside of *VANDAL21* loci within the genome. A specific feature in the regions with VANC21 localization is that multiple motifs are arrayed in the same orientation at high density (Fig. 2b; Supplementary Fig. 6). The high density of the YAGTATTAY motif was unique to the *VANDAL21* loci (Supplementary Fig. 8b), and a low density of the YAGTATTAY motif outside *VANDAL21* was not associated with VANC21 localization. These results suggest that VANC21 recognizes "YAGTATTAY" motifs and that motif density is important for the specific chromosomal localization of VANC21.

**VANC targets evolve through tandem repeat formation**. The *VANDAL21* family can be classified into two subgroups based on sequence similarity (Fig. 3d; Supplementary Fig. 8c). They are shown as *VANDAL21_1* and *VANDAL21_2* in Fig. 3d. *VAN-DAL21_1* is the group with the autonomously-mobile copy *Hi*. We noticed that while "YAGTATTAC" motifs (hereafter called C-type) were found in both subfamilies, "YAGTATTAT" motifs (hereafter called T-type) accumulated only in *VANDAL21_2*. Although *VANDAL21_2* copies exist in *A. lyrata*, these copies do not possess the T-type motif (Fig. 3d; *A. lyrata*-specific lineages are shown by red). These results suggest that gain or loss of multiple motifs occurred relatively recently and occurred even within *VANDAL21* members.

We then wondered how the multiple motifs have accumulated so rapidly. Separation between *A. thaliana* and *A. lyrata* has been estimated to be 5–10 million years[21,22], and average base substitution rate between these two species for neutral sites has been estimated to be 0.13[23]. It is hard to account for the accumulation of multiple motifs by simple base substitutions.

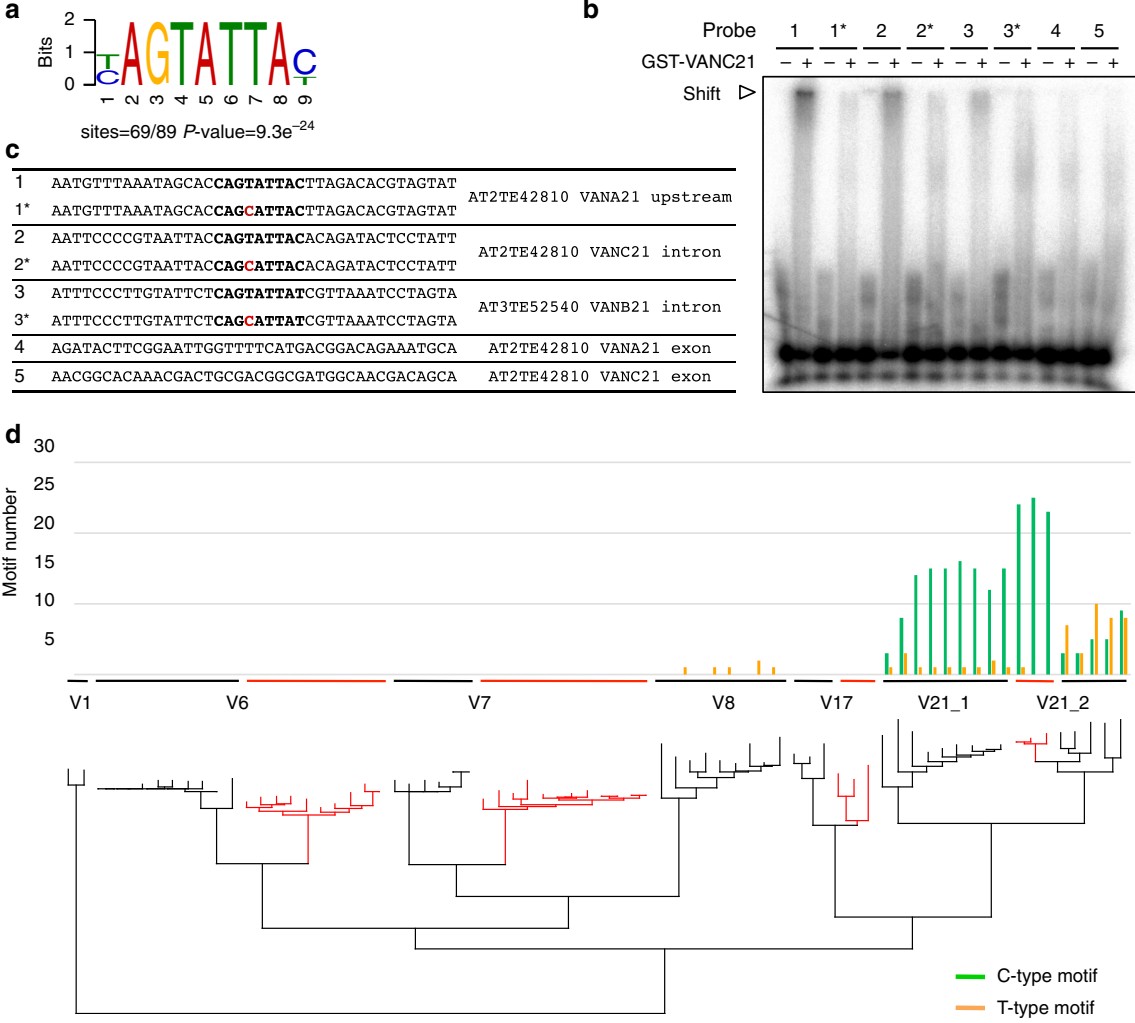

**Fig. 3** VANC21-binding motifs and their distribution among *VANDAL* families. **a** DNA sequence motif most commonly found at FLAG-VANC21-binding sites in *VANDAL21* TEs. Localizations of the motif are shown in Fig. 2b and Supplementary Fig. 6, with green and orange bars indicating positions of the C-type ("YAGTATTAC") and T-type ("YAGTATTAT") motifs, respectively. **b** Electrophoretic mobility shift assay (EMSA) by VANC21 protein for double-stranded DNA of the sequences shown in **c**. The probe sites are also shown by white arrowheads in Fig. 2b. 1*, 2*, and 3* have single-base substitutions within the motif relative to the original sequences. 4 and 5 are control sequences from the exons, where VANC21 localization was not detected. Results for dependence of the shift to the protein amount and a competition assay are shown in Supplementary Fig. 7. **c** Sequences of the dsDNA probe used in EMSA (**b**). **d** Numbers of C- and T-type motifs within *VANDAL21* and related *VANDAL* family members within the genomes of *A. thaliana* and *A. lyrata*. *A. lyrata*-specific lineages are shown with red lines. IDs for these TE copies and bootstrap probabilities are shown in Supplementary Fig. 8c

Interestingly, regions with the motifs are often part of tandem repeats, and comparison of these sequences among different *VANDAL21* copies revealed extensive gain and/or loss of the repeats in related copies (Fig. 4a–f; Supplementary Fig. 9). Furthermore, multiple units of sequences within a repeat often diverged together. For example, as illustrated in Fig. 4d, copy #2 has six motifs while copy #3 only has two motifs even though they both have a tandem repeat structure of similar organization. This pattern suggests that multiple copies within the repeat evolved together. These features were found in the introns of *VANC21* and *VANB21*, as well as in upstream regions of *VANA21* (Fig. 4b–f), suggesting that these tandem repeats are very dynamic.

As shown in Fig. 1, VANC proteins have also evolved to change their target specificities. Interestingly, VANC proteins have conserved domains and highly variable regions; and the variable regions often contain tandemly arrayed peptide motifs (Fig. 5a; Supplementary Fig. 10). In addition, *VANC* genes have diverse exon/intron organizations, which are frequently

associated with tandem repeat formation (Fig. 5b). We speculate that these tandem repeats within VANC proteins might play roles in defining target specificity, as is the case for the tandem repeats in their targets.

**Sequences affected by VANC6 have different motifs.** We also characterized the targets of VANC6. As VANC21 localization is associated with local loss of CG methylation (Fig. 2), we predicted targets of VANC6 by loss of CG methylation in transgenic lines expressing *VANC6* (Supplementary Table 2). Motifs identified, "AGTTGTCC (CC-type)" and "AGTTGTAC (AC-type)", are at least two nucleotides different from target motifs of VANC21 (Supplementary Fig. 11; Supplementary Table 2). These motifs are enriched in *VANDAL6* and related *VANDAL* family members, such as *VANDAL8* (Supplementary Fig. 11c), which show hypomethylation in *VANC6* transgenic lines (Supplementary Fig. 3). In addition, VANC6 protein bound to a region with high density of the motifs in vitro (Supplementary Fig. 12).

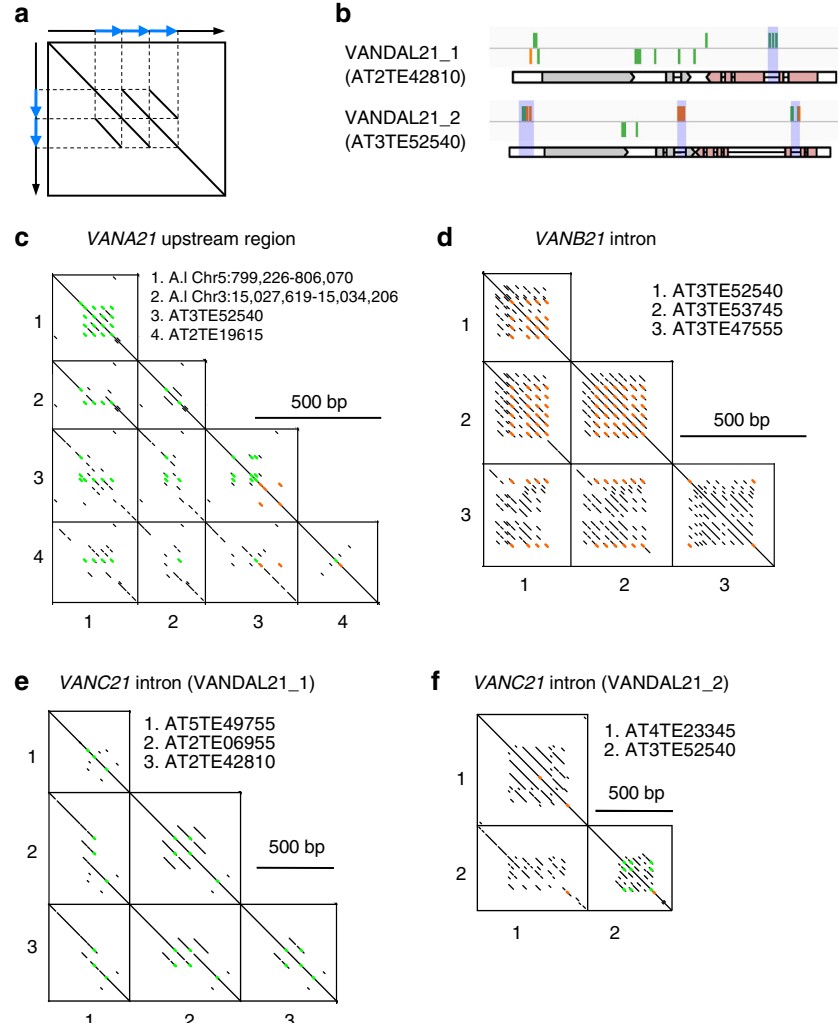

**Fig. 4** Evolution of VANC21-binding regions. **a** Schematic diagram of dot plot for sequences with tandem repeat structures. Blue arrows indicate tandem repeats. Because of sequence identity, tandem repeat structure generates parallel lines. Difference in copy number can also be detected; in this case, three and two repetitions for *X*- and *Y*-axes, respectively. **b** Structures of *VANDAL21* copies, with regions analyzed in **c–f** shown with shadow. Green and orange bars show C-type and T-type motifs. **c–f** Dot plots comparing the *VANDAL21* sequences upstream of *VANA21* (**c**), *VANB21* intron (**d**), and *VANC21* intron (**e**, **f**). Regions with 10 bp exact match are shown by dots. Green and orange indicate regions with C- and T-type motifs, respectively. Sequence alignments in these tandem repeat regions are shown in Supplementary Fig. 9

Interestingly, these motifs recognized by VANC6 are also found in *AT9TSD1*, which show hypomethylation in *VANC6* transgenic lines (Supplementary Fig. 4; Supplementary Data 1), even though *AT9TSD1* is not related to *VANDAL*. In addition, some of *AT9TSD1* are transcriptionally derepressed in *VANC6* transgenic lines (Supplementary Fig. 4). These results support our interpretation that each VANC protein induces loss of DNA methylation and transcriptional derepression by recognizing specific local sequences. The CC- and AC-type motifs are found as tandem repeats in non-coding regions of corresponding *VANDAL* copies, as is the case for motifs recognized by VANC21 (Supplementary Fig. 13). Thus, despite having a completely different spectra of target TEs (Fig. 1g; Supplementary Fig. 1f–h), VANC6 appears to function in a manner similar to that of VANC21 and their target sequences have similar evolutionary dynamics.

## Discussion

In this report, we show that VANC proteins have high specificity for their target sequences and evolve together with their targets. The differentiated target regions of VANCs accumulated recognized motifs in the form of tandem repeats (Fig. 4). Theoretically, tandem repeats can expand and contract by replication slippage and/or unequal crossing-over[24]. The evolution of tandem repeats can occur rapidly, as is the case for centromeric tandem repeats in animals and plants[25,26]. In the case of centromeric tandem repeats, centromeric histone H3, a protein localizing in the repeat, also evolves rapidly in its N-terminal domain, although the C-terminal core region is conserved[27,28]. Interestingly, VANC proteins also have conserved domains and highly variable regions; and the variable regions are frequently associated with tandem repeat structures (Fig. 5). Investigation of regions within VANC proteins defining target specificity will be a focus of future studies.

Tandem repeats are often a target of epigenetic silencing. For instance, tandem repeats were formed multiple times independently during evolution of *FWA* gene promoters in the genus *Arabidopsis*[29]. *FWA* is an imprinted gene and its promoter is a target of active DNA demethylation by DEMETER protein[30,31]. It is tempting to speculate that anti-silencing by VANC also increases the frequency of tandem repeat formation during evolution. Tandem repeats can also be target of RNAi machinery[32,33]. Evolution

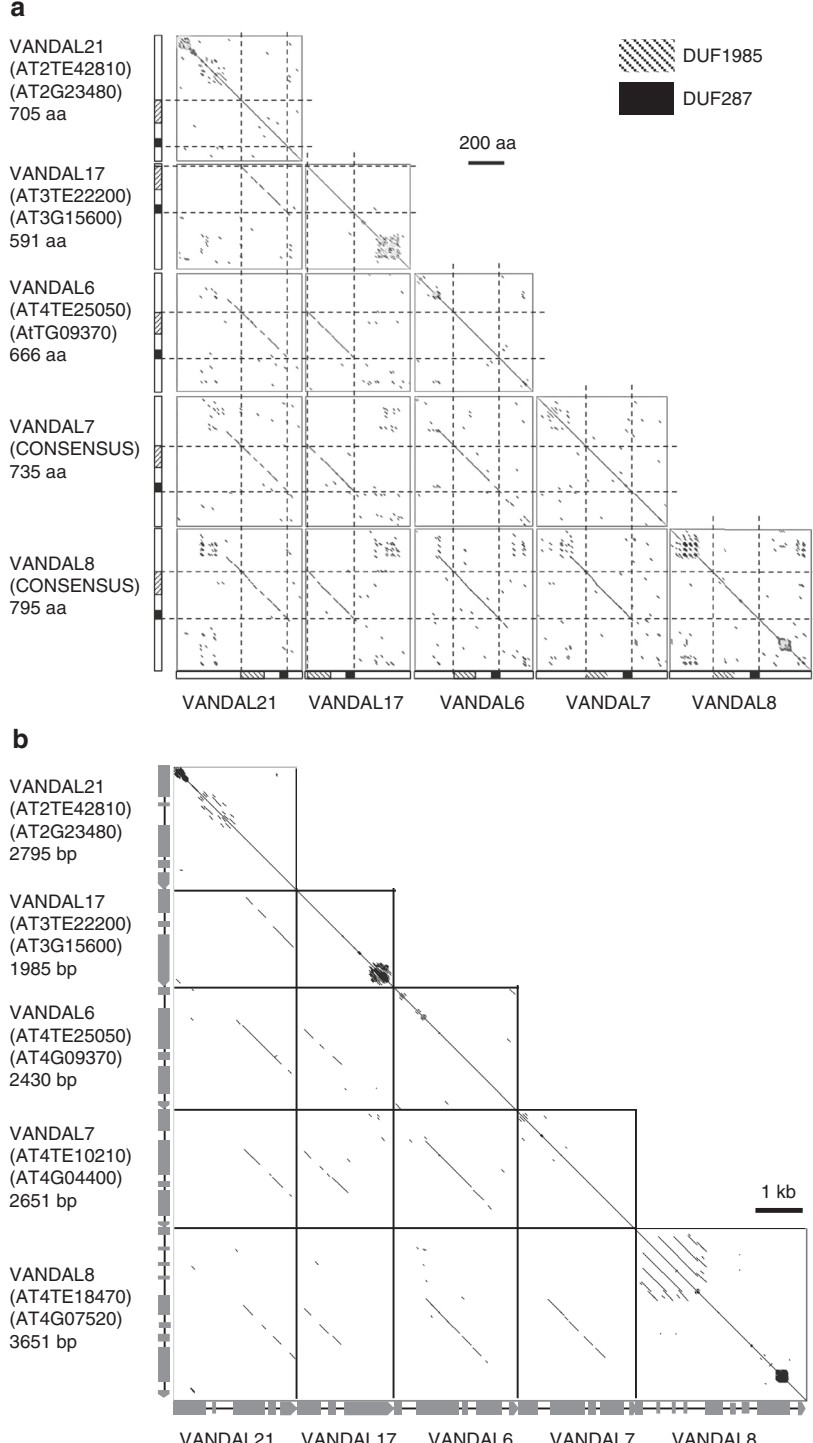

**Fig. 5** Dot plot analyses of the VANC proteins. **a** Comparison of amino-acid sequences of the VANC proteins. Homologous regions were plotted with dotmatcher program (window size: 10, threshold: 23). Amino-acid sequences (N′–C′) were ordered from top to bottom and left to right. Scale bar for 200 a.a. was shown in the right of plots. Two domains (DUF1985 and DUF287) are shown by shaded and filled areas, respectively. VANA (transposase), another protein encoded in these VANDAL members, is much more conserved (Supplementary Fig. 10). **b** Comparison of the nucleotide sequences of *VANC* genes. *VANC* genes were plotted with YASS program under default parameters[52]. DNA sequences are ordered from top to bottom and left to right. Gray boxes indicate exons. Divergence between these *VANDAL* families as well as divergence between the *VANDAL* copies in *A. thaliana* and *A. lyrata* is shown in Supplementary Tables 3, 4 and Supplementary Fig. 14

of motifs targeted by RNAi may offer a short-term advantage to escape from the host defense system. Such a short-term advantage may lead to a long-term advantage in allowing proliferation while causing minimal damage to the host, because differentiation of anti-silencing systems would limit the number of proliferating TEs.

An important remaining question is how the anti-silencing is achieved after the sequence-specific binding of VANCs. One possible pathway could be that VANC primarily functions as a transcription activator and that transcription induces the loss of silent marks. However, VANC21 induced hypomethylation not

only in coding regions but also in intergenic regions. In addition, hypomethylation was also found in the *VANDAL* copies without detectable transcriptional activation (Supplementary Figs. 4, 6), suggesting that the primary effect of VANC21 is not transcriptional activation but removal of silent mark(s). We are currently trying to establish systems to genetically screen for mutants affecting VANC-mediated anti-silencing, in order to identify host factors involved in this process.

As our results demonstrate that a combination of short motifs and the VANC proteins can induce anti-silencing in diverse sequences, they could potentially be used to engineer epigenetic states at specific loci. For example, the anti-silencing may be used to ensure expression of transgenes in genetically modified crops[34,35]. It would also be interesting to learn if some of the host machinery for the sequence-specific anti-silencing is conserved in other kingdoms.

## Methods

**Plant materials.** *A. thaliana* strain Columbia-0 (Col-0) was used as "wild type". Transgenic lines with full-length *Hi* and ∆*AB Hi* in pPZP2H-lac were described previously[18]. The VANC6 construct was generated by two rounds of PCR from genomic DNA and cloned into pPZP2H-lac vector after digestion by *Spe*I and *Xho*I. Primer sequences for this and other constructions are available upon request. The FLAG-tagged VANC21 construct was generated by two steps: (i) For generating FLAG-VANC21 construct in which 3x FLAG tag was fused at C-terminus VANC21 CDS, 3x FLAG sequence (5′-GACTACAAAGACGATGACGA-CAAGGATTATAAGGATGACGATGATAAAGACTATAAAGATGATGATGA-CAAA-3′) and linear ∆*AB Hi* in pBluescript II SK (−) were generated by PCR and they were combined using In-Fusion HD cloning kit (Takara); (ii) The FLAG-tagged VANC21 sequence was PCR amplified and cloned in *Sma*I-digested pPZP2H-lac vector using an In-Fusion HD cloning kit.

**Whole-genome bisulfite sequencing.** Mature rosette leaves were used for genomic DNA extraction. Bisulfite treatments and library preparations were performed as described previously[18]. Paired-end reads were qualified using Trimmomatic-0.33 software with following options "ILLUMINACLIP:TruSeq3-PE.fa:2:30:10 LEADING:3 TRAILING:3 SLIDINGWINDOW:4:15 MINLEN:36"[36]. Qualified reads were mapped using the "bismark" command of bismark (0.14.3) software with following options "-n 1 -l 20". PCR duplicates were removed from mapped bam files by "deduplicate_bismark" command[37]. Base resolution of read counts of methylated and unmethylated cytosines were obtained as CX_reports files by "bismark_methylation_extractor" command with following options "--bedGraph --CX --cytosine_report". Reads from previous study were used for the wild-type data[38]. Differentially methylated regions at CG sites (CG-DMRs) induced by *VANC* genes were defined as previously described[38]. Briefly, in each 100-bp window, DMRs were defined when a difference of methylation level at CG sites was 0.5 or more. Multiple DMRs were merged if they were adjacent to each other or there was only one gap of the 100-bp window. DNA sequences of CG-DMRs at *VANDAL21* TEs induced by ∆*AB* ($N = 93$), and CG-DMRs at *VANDAL6*, *7*, *8*, *17*, and *AT9TSD1* TEs induced by VANC6 ($N = 89$), were used for identifying statistically enriched short motifs, respectively (described below).

**ChIP-seq.** ChIP was performed according to the methods reported by Gendrel et al.[39] but with modification. Approximately 5.0 g of mature rosette leaves was fixed with 1% of formaldehyde. The fixed leaves were ground in liquid nitrogen, resuspended with 50 ml of extraction buffer 1 (0.4 M sucrose, 10 mM Tris-HCl pH 8, 5 mM β-mercaptoethanol, cOmplete, EDTA-free protease Inhibitor Cocktail (hereafter cOmplete; Sigma-Aldrich)), and the solution was filtrated with two layers of Miracloth (Millipore). The filtrated solution was centrifuged (1900×*g*, 20 min) and the precipitate was washed twice with 5 ml of extraction buffer 2 (0.25 M sucrose, 10 mM Tris-HCl (pH 8.0), 10 mM MgCl₂, 1% Triton X-100, 5 mM 2-mercaptoethanol, cOmplete). The precipitate was resuspended with extraction buffer 3 (1.7 M sucrose, 10 mM Tris-HCl, 0.15% Triton X-100, 2 mM MgCl₂, cOmplete), layered on 500 µl of extraction buffer 3, and centrifuged (15,000×*g*, 40 min). The precipitate containing chromatin was resuspended with 500 µl of nuclei lysis buffer (50 mM Tris-HCl (pH 8.0), 10 mM EDTA, 1% SDS, cOmplete). Chromatin was sheared by Branson Sonifier 250D with the conditions of Duty 17%, Pulse 60 s, 15 times. After centrifugation, 100 µl of supernatant was diluted with ChIP Dilution buffer (1.1% Triton X-100, 1.2 mM EDTA, 16.7 mM Tris-HCl (pH 8.0), 167 mM NaCl) into 1000 µl, and incubated with either 10 µl of antiserum of 6xHis-VANC21-immunized rabbit, or 7 µg of anti-FLAG antibody (F7425 Sigma-Aldrich) overnight. Immune complexes were captured by incubating with 100 µl of Dynabeads Protein G (Thermo Fisher Scientific) for 1 h. Above procedures were performed at 4 °C. After rinsing the magnetic beads, immunoprecipitated DNA–protein complexes were eluted and reverse-crosslinked by incubation of the beads with 200 µl of direct elution buffer (10 mM Tris-HCl (pH 8.0), 0.2 M

NaCl, 5 mM EDTA, 0.5% SDS) for overnight at 65 °C. Amount of DNA was quantified with the Qubit dsDNA High Sensitivity Assay Kit (Thermo Fisher Scientific). About 1.2 ng of DNA was used for library construction using a KAPA hyper prep kit (Kapa Biosystems) following manufacturer's protocol. The libraries were amplified by 15 cycles of PCR using KAPA Hifi-PCR solution, and sequenced either by Miseq as 74 bp of paired-end reads or Hiseq 4000 as 50 bp of single-end reads. Reads were mapped by Bowtie (0.12.8)[40]. For paired-end reads, "-X 1000" option was used. For single-end reads, "-n 2 −M 1 --best" option was used, because reads derived from repetitive regions were often not mapped uniquely and this option allows non-unique reads to be mapped on a region selected randomly from multiple best hits. Resulting sam files were converted into bam files and sorted by SAMtools (0.1.18)[41]. To identify peaks of FLAG-VANC21, a sorted bam file of anti-FLAG immunoprecipitated sample of FLAG-VANC21 transgenic plants was analyzed using the MACS2 (2.1.0) "callpeak" command with the options "-g 135000000 -B -q 0.01"[42]. DNA sequences of VANC21-enriched regions defined by MACS2 in VANDAL21 TEs were extracted. Short motifs that were statistically enriched at the VANC21-binding regions in VANDAL21 TEs were searched by a DREME script of MEME software (4.11.0) under default parameters except maximum core width was set as 9[43]. Same parameters were used for searching short motifs at CG-DMRs induced by *VANC* genes (Supplementary Table 2). In the plots of Fig. 2a and Supplementary Fig. 5d–g, regions with centromeric satellite repeats are excluded, because those regions show unstable signals for both IP and input samples. Reads mapped on specific regions were counted by "coverage" command of BEDtools (2.16.2)[44]. These data sets were visualized on IGV genome browser[45]. TAIR10 annotation was used for all sequence analyses.

**Generation of anti-VANC21 polyclonal antibody.** Total RNA was isolated from *Hi* transgenic plants by the PureLink Plant RNA Reagent (Thermo Fisher Scientific). About 1 µg of RNA was used for complimentary DNA (cDNA) synthesis with AMV ver3.0 (Takara). *VANC21* cDNA was amplified by PrimeSTAR GXL (98 °C 10 s, 60 °C 15 s, 68 °C 3 min; 30 cycles) and A-tailed by ExTaq (Takara). The cDNA was TA-cloned into pGEM T-easy vector (Promega) by Mighty Mix (Takara). Cloned full length of *VANC21* cDNA was amplified with PrimeSTAR GXL and primers with AttB1 and AttB2 sequences. The PCR fragment was cloned into pDEST17 vector by one-tube BP and LR Gateway reaction system following manufacturer's protocol (Thermo Fisher Scientific). The pDEST17 vector containing *VANC21* cDNA was transformed into *Escherichia coli* of BL21-Al strain. Cells were pre-cultured for 8 h in 5 ml of LB liquid medium and the 0.5 ml of the culture was inoculated in 25 ml of LB liquid medium. After 3 h of incubation, expression of 6xHis-tagged VANC21 protein (6xHis-VANC21) was induced for 3 h by adding up to 0.2% of L-arabinose. All culture steps were performed at 37 °C. Cells were harvested by centrifugation at 6000×*g* for 10 min. Cells were lysed, and the insoluble fraction containing 6xHis-VANC21 was purified using Bugbuster Master Mix (Millipore) following manufacturer's protocol. The purified insoluble fraction was solubilized in denaturing binding buffer (6 M urea, 30 mM imidazole, 1× PBS buffer). 6xHis-VANC21 was captured by HisTrap Ni sepharose column (GE), and eluted with elution buffer (6 M urea, 200 mM imidazole, 1× PBS buffer). Purified 6xHis-VANC21 was used for immunizing rabbits (MBL).

**Western blotting.** To extract nuclear proteins without fixation, 1.2 g of frozen mature leaves was ground and resuspended in 25 ml of nuclear extraction buffer 1 (0.4 M sucrose, 10 mM Tris-HCl (pH 8.0), 10 mM MgCl₂, 5 mM β-mercaptoethanol, and cOmplete). Extract was filtered through Miracloth (Millipore) and incubated on ice for 20 min. After centrifuging at 2000×*g* for 10 min, supernatant was discarded and pellet was resuspended in 1 ml of nuclear extraction buffer 2 (0.25 M sucrose, 10 mM Tris-HCl (pH 8.0), 10 mM MgCl₂, 1% Triton X-100, 5 mM β-mercaptoethanol, cOmplete). This washing step was repeated three times. Supernatant was discarded and pellet was resuspended in 350 µl of lysis buffer (50 mM Tris-HCl (pH 8.0), 10 mM EDTA, 1% SDS, cOmplete) and sonicated three cycles of 10 s, followed by 50 s incubation on ice. After centrifuging at 12,000 rpm for 10 min, supernatant was mixed with SDS sample buffer. After heating at 95 °C for 5 min, protein was separated by SDS–PAGE (8%) and transferred to PVDF membrane (GE) by wet condition in transfer buffer (192 mM glycine, 25 mM Tris, 15% methanol). After the transfer, the membrane was immuno-hybridized with the following steps: blocking in TBS-T buffer (50 mM Tris-HCl, 150 mM NaCl, 0.05% Tween 20) containing 0.5% of skim milk (Nacalai Tesque) for 1 h, washed with TBS-T buffer three times, incubated in Can Get Signal Solution I (TOYOBO) containing 1:1000 dilution of either anti-VANC21 antiserum or anti-FLAG antibody (F7425 Sigma-Aldrich) for 1 h, washed with TBS-T buffer three times, incubated in Can Get Signal Solution II containing 1:10,000 dilution of anti-rabbit HRP-labeled secondary antibody (074-1506 KPL) for 1 h, and washed with TBS-T buffer three times. ECL prime western blotting detection reagents (GE) was used to induce chemiluminescence. Signals were analyzed by LAS4000mini (GE).

**RNA-seq.** RNA was extracted from mature rosette leaves of WT or ∆*AB Hiun* plants by PureLink Plant RNA Reagent (Thermo Fisher Scientific) and sent to Takara Biomedical Center (Takara) for strand-specific library preparation and sequencing. For RNA-seq, paired-end reads were mapped by tophat (2.1.0) with

the following parameters[46] "--library-type fr-firststrand --mate-inner-dist 200 --max-multihits 1".

**EMSA**. VANC proteins were synthesized in *E. coli*. For *VANC6*, total RNA was isolated from *ddm1-1* plants by TRIzol (Thermo Fisher Scientific). About 1 μg of RNA was used for cDNA synthesis with PrimeScript RT-PCR Kit (Takara) using random 6-mer primers. *VANC6* cDNA was amplified by Phusion (NEB) (98 °C 5 s, 58 °C 10 s, 72 °C 1 min; 30 cycles) and A-tailed by ExTaq (Takara). The cDNA was TA-cloned into pGEM T-easy vector (Promega). The cDNA of *VANC21* and *VANC6* was cloned into pDEST15 vector as described above. These vectors were then transformed into *E. coli* of BL21-Al strain. Cells were pre-cultured for 8 h in 5 ml of LB liquid medium and 1 ml of the culture was inoculated in 100 ml of LB liquid medium. After 3 h of incubation at 37 °C, the culture was incubated at 25 °C for 1 h. Expression of N-terminus-GST-tagged VANC21 protein (GST-VANC21) was induced for 24 h at 25 °C by adding up to 0.2% of L-arabinose. Cells were harvested and lysed by BugBuster Master Mix. The lysed solution was diluted with PBS buffer up to 50 ml and incubated with 0.125 g of glutathione–agarose beads (Sigma-Aldrich, equilibrated with PBS buffer to 1.4 ml) for overnight at 4 °C. The beads were washed with PBS buffer four times and the GST-VANC21 protein was eluted in buffer containing 10 mM glutathione and 50 mM Tris-HCl (pH 7.5). The protein concentration was quantified using a Nanodrop 2000 (Thermo Fisher Scientific). About 10 pmol of double-stranded DNAs (dsDNAs) was radiolabeled with 1 U of T4PNK (Takara) by incubation with 0.5 MBq of [γ-32P] ATP for 1 h at 37 °C in 10 μl of reaction solution, and the radiolabeled dsDNAs were column purified using MicroSpin G-25 Columns (GE). About 2.5 μg of GST-VANC21 was incubated at 4 °C for 30 min in 15 μl of reaction solutions with 0.15 pmol of radiolabeled dsDNA in buffer containing 15 mM Tris-HCl (pH 7.5), 300 mM NaCl, 3 mM MgCl₂, 0.04% Triton X, 4% glycerol, 0.5 mM DTT, and 30 ng of Poly: IC. The reaction solutions were separated on 6% non-denaturing polyacrylamide gel with 1× TBE buffer by 350 V for 15 min. Radioactivity signals were detected using a FLA-9000 (FUJI). In EMSA in Supplementary Fig. 12, VANC proteins without GST tag were used. After cloning VANC cDNAs into pDEST15 vector, PreScission Protease (GE) recognition sequence was added by inverse PCR. VANC proteins were purified with same procedures described above, except for elution with PreScission Protease (GE).

**Phylogenetic analyses**. Estimation of phylogeny of *VANDAL* families was performed as described previously[18], except that sequences were aligned by MUSCLE algorithm, and ClustalX (2.0.12) was used for constructing neighbor-joining tree[47]–[50]. Alignments of non-coding regions were also performed using the MUSCLE algorithm.

**Dot plot analyses of VANA and VANC proteins**. The CDS regions were used for the analyses. CDS regions were obtained based on the TAIR10 annotation for *VANDAL17*, *VANDAL21*, and *VANDAL6*. For *VANDAL7* and *VANDAL8* families, consensus sequences were generated and used for the analyses, because most members have become pseudogene. For *VANDAL8*, VANC annotations differ between loci due to several indel variations in the possible exon regions. Thus exon–intron junctions were determined based on the cDNA sequence data. Dot plots were made by using amino-acid sequences with EMBOSS dotmatcher default setting (10 window size, 23 threshold).

**Estimation of genetic distance**. Genetic distances between species and between families of *VANDAL* TEs were estimated by Poisson correction distances. Amino-acid sequences of conserved domains of VANA and VANC were used for the alignment. VANA (transposase) sequences used for calculating genetic distances were as described previously[18]. For *VANC* gene, exon–intron structure of *A. thaliana* copies was used for CDS structure in *A. lyrata* and used for between species comparisons only. Number of tandemly repeated regions was estimated by using Tandem repeat finder program[51]. Sequences used were retrieved from TAIR database, of which we used only copies with both *VANA* and *VANC* genes annotated in the TE and with VANC having complete structure of CDS. Full-length CDSs of VANC were then used for the analyses. When multiple patterns of repeat structure can be identified in the overlapping region, we only used that covering the longest region.

**Data availability**. The sequence data were deposited into DDBJ (WGBS, ChIP-seq, and RNA-seq data as DRA006000, DRA006001, and DRA006002, respectively). The authors declare that all other data supporting the findings of this study are available within the article and its Supplementary Information files, or from the corresponding authors on request.

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

## Acknowledgements

We thank Kae Kato and Akiko Terui for technical assistance, Hiroyuki Araki, Kenichi Nonomura, Naruya Saito, and Taiko To for technical advice, Yasushi Hiromi, Hideki Innan, Damon Lisch, and Eric Richards for comments on the manuscript. This work used the Vincent J. Coates Genomics Sequencing Laboratory at UC Berkeley, supported by NIH S10 OD018174 Instrumentation Grant. Computations were partially performed on the NIG supercomputer at NIG, Japan. Supported by grants from Mitsubishi Foundation (to T.K.), Japanese Ministry of Education, Culture, Sports, Science and Technology (26221105 and 15H05963, to T.K.), JST CREST Grant, Japan (JPMJCR15O1 to T.K.), and Systems Functional Genetics Project of the Transdisciplinary Research Integration Center, ROIS, Japan (to A.T., A.F., Y.T., and T.K.).

## Author contributions

A.H., R.S., K.T., T.S., Y.F., A.T., A.F. and Y.T. contributed to the experiments. A.H., R.S., T.S., A.K. and T.I. contributed to data analyses. A.H. and T.K. contributed to writing the paper.

## Additional information

**Competing interests:** The authors declare no competing financial interests.

