## [Peer Review File · Nature Communications]

Reviewers' comments:

Reviewer #1 (Remarks to the Author):

Earlier work in maize has revealed that some transposable elements (e.g. MuDR and Spm) have mechanisms to counteract DNA methylation. In this manuscript, Hosaka et al use Arabidopsis transgenic constructs and genome-wide approaches to show that VANC proteins can specifically bind transposable elements of the VANDAL family by which they are encoded to induce DNA demethylation through a still unknown process.

Previous work from the same group demonstrated that introduction of a Hi transgene in which only the VANC ORF remains is sufficient to induce excision of endogenous Hi and demethylation of its 3' end (Fu et al. 2013). The description of the methylation defects induced by VANC21 expression is extended in the current manuscript and the authors show that a VANC protein encoded by a distinct VANDAL family can also induce demethylation of its corresponding family members. Focusing on VANDAL21 elements and the VANC21 protein, and comparing ChIP-seq and BS-seq data, the authors show that DNA demethylation is prominent at VANC21 binding sites, which are enriched in non-coding regions of VANDAL21 copies. By analyzing DNA regions where VANC21 localizes, the authors identify a short-repeated DNA sequence motif and use gel-shift assays to demonstrate that VANC21 can efficiently and specifically bind this motif. The authors found that copy number of these DNA repeats is dynamic and varies between VANDAL21 family members, suggesting rapid evolution. Additionally, VANC proteins contain repeated peptide motifs in their non-conserved regions that also seem to evolve rapidly.

Even though the molecular mechanism by which binding of VANC proteins onto VANDAL copies lead to loss of DNA methylation remains unknown, the work reported here should be of interest to those interested in transposable elements and silencing, as well as to evolutionary biologists. The experiments are well designed and executed, and most of the conclusions are fully supported. Addressing the following points should help clarifying some mechanistic aspects.

Major points:

1) The authors provide an extensive description of the VANC21 / VANDAL21 interactions. However, it would be informative to extend the analysis to at least another VANC/VANDAL combination (eg. VANC6/VANDAL6) to strengthen the conclusions about specificity of the whole system. For instance, while VANC21 specifically induces DNA demethylation at VANDAL21 copies, data in figure 1e show that expression of VANC6 affects methylation levels not only at VANDAL6 but also at VANDAL8 elements. The authors should provide methylation data for all VANDAL families in WT, Δ AB TG and VANC6 TG backgrounds. Can repeated DNA motifs be detected in non-coding regions of VANDAL6 members, and if yes, how divergent are these motifs from those present in VANDAL21? Although VANC21 appears to specifically bind the repeated DNA motif present in VANDAL21 elements, the authors should assess whether or not VANC6 can bind VANDAL21 DNA repeats.

2) Coevolution between the DNA repeats and the VANC proteins is not strongly supported. Importantly, it is not clear from the current data if the repeated peptide motifs present in VANC21 (and other VANC proteins) are required for VANC21 binding to DNA repeats. Analyzing binding properties of VANC21 versions with deleted/mutated peptidic repeats would be informative.

Minor points:

1) Panels c, d, f, g of Figure 1 are not trivial to understand. The authors may consider using another way of illustrating the data.

2) Numbered arrow heads on Figure 2b are not explained in the corresponding legend.

- 3) Expressing VANC21 results in release of silencing at endogenous VANDAL21 elements. Does expression of VANC6 also lead to loss of silencing at VANDAL6 elements?
- 4) In the discussion section, the authors state that rapid evolution of DNA repeats may have short-term advantage to escape from the host RNAi defense system. Although this is easy to conceive, how this would lead to "the long-term advantage to proliferate with minimum damage to the host" is less clear.

Reviewer #2 (Remarks to the Author):

The manuscript submitted by Hosaka et al. describes the diversity and in some extent the evolutionary dynamics of "anti-silencing" TE-related sequences in Arabidopsis. VANC, one of the 3 ORF of the transposon has been shown to be involved in the silencing process by the same group in a previous publication. One open question concerns the specificity of VANC with Hi transposons, as well as the mechanisms of anti-silencing.

The authors first show that VANC protein is necessary for the demethylation of Hi transposons. They also show that the specificity is found in fast evolving regions of the TE sequence. These are tandem repeats, known to be highly dynamic.

I have no major objection for the publication of this manuscript in Nature communication. My only problem is that the manuscript lacks some estimates of the evolutionary rates :

The statement of "rapid evolution" is found in several instances throughout the manuscript and is not supported by precise figures. What is rapid ? for instance, the authors use the phylogeny of Vandal elements to show that the T-type motif is only found in *A. thaliana* and absent from *A. lyrata*. What is the divergent time between *A. thaliana* and *A. lyrata* ? Could the authors provide an estimate for the occurrence of T-motif ?

Similarly, the comparisons of tandem arrays among VANC paralogs in *A.t* genomes show the frequent occurrence of VNTRs. The authors could estimate this frequency : 1 - by aligning paralogs sequence of complete elements 2 - compute divergence 3 - apply molecular clock to provide an estimation of VANDAL amplification in the genome, which would provide a time frame for diversification of tandem arrays.

minor points :

- introduction : TE HTs are not rare in eukaryotes. See Peccoud et al., PNAS 2017 and ElBaidouri et al, Genome Research 2014.
- replace genera Arabidopsis by Arabidopsis genus.

Point-by-point response to the reviewers' comments (response shown by blue):

Reviewer #1

This reviewer is very positive. Nonetheless, the comments are very constructive. The manuscript has been much improved by incorporating comments of this reviewer.

Major points

1) The authors provide an extensive description of the VANC21 / VANDAL21 interactions. However, it would be informative to extend the analysis to at least another VANC/VANDAL combination (eg. VANC6/VANDAL6) to strengthen the conclusions about specificity of the whole system. For instance, while VANC21 specifically induces DNA demethylation at VANDAL21 copies, data in figure 1e show that expression of VANC6 affects methylation levels not only at VANDAL6 but also at VANDAL8 elements. The authors should provide methylation data for all VANDAL families in WT, Δ AB TG and VANC6 TG backgrounds.

Our response:

As suggested, we showed methylation data for all VANDAL families in WT, Δ AB TG and VANC6 TG backgrounds (Supplementary Fig. 3). As expected from Fig 1, Δ AB affects *VANDAL21*, while VANC6 TG affects not only *VANDAL6* but also *VANDAL8*, which is related to *VANDAL6*. In addition, we showed methylation data of other TEs in WT, Δ AB and VANC6 TG (Supplementary Data 1). VANC6 TG also induced weak hypomethylation in *AT9TSD1*, a family of Mutator-Like Element, unrelated to *VANDAL* (Supplementary Fig. 4, bottom part). The effects of VANC6 on *VANDAL8* and *AT9TSD1* can be understood by importance of specific local sequence motifs for targeting VANCs (shown in our next response below).

Can repeated DNA motifs be detected in non-coding regions of VANDAL6 members, and if yes, how divergent are these motifs from those present in VANDAL21?

Our response:

As VANC21 localization is associated with local loss of CG methylation (Fig. 2), we predicted targets of VANC6 by loss of CG methylation (Supplementary Table 2). Motifs identified, “AGTTGTCC” and “AGTTGTAC”, are at least two nucleotides different from those found for VANC21 binding (Supplementary Fig. 11d), and they are found in *VANDAL6* and *VANDAL8* copies (Supplementary Fig. 11). Very interestingly, these motifs are also found in *AT9TSD1* (a family of Mutator-Like Element, unrelated to *VANDAL6*, Supplementary Fig. 4), which show hypomethylation in VANC6 transgenic lines (Supplementary Fig. 4, Supplementary Data 1). These results strongly suggest that VANC6 recognizes specific local sequences to induce loss of DNA methylation, as is the case in VANC21.

Although VANC21 appears to specifically bind the repeated DNA motif present in VANDAL21 elements, the authors should assess whether or not VANC6 can bind VANDAL21 DNA repeats.

Our response:

As suggested, we examined binding of VANC6 to target DNA of VANC21. As expected, VANC6 did not bind target DNA for VANC21 (Supplementary Fig. 12 left half). In addition, we compared binding of VANC6 and VANC21 to a hypomethylation target of VANC6 (Supplementary Fig. 12 right half). VANC6, but not VANC21, binds to the VANC6 target. These in vitro results are fully consistent with the in vivo results for the differentiated sequence-specific loss of DNA methylation by VANC21 and VANC6, further strengthening our proposal for the differentiation of VANC proteins and target sequences by direct recognition of short target sequences.

2) Coevolution between the DNA repeats and the VANC proteins is not strongly supported. Importantly, it is not clear from the current data if the repeated peptide motifs present in VANC21 (and other VANC proteins) are required for VANC21 binding to DNA repeats. Analyzing binding properties of VANC21 versions with deleted/mutated peptidic repeats would be informative.

Our response:

In regard to the evolution of binding ability of VANC proteins, the most critical question would be how the target sequence specificity is determined. We regard so because rapidly evolving is the sequence specificity, rather than the binding activity in general (Supplementary Fig. 12). Currently we are starting to make chimeric constructs of VANC21 and VANC6 to dissect regions defining the target sequence specificity. We are also starting alanine scan for the protein. We do have preliminary results for VANC21 deletion constructs, some of which did not bind the VANC21 target DNA in vitro. However, it is difficult to clearly distinguish between loss of sequence specificity and loss of binding ability itself. In addition, deletion could disrupt protein conformation, mimicking loss of function in other regions. Given preliminary nature of the results of the deletion experiments, we would like to refrain from including those results in this manuscript. We would rather like to integrate these results into more thorough analyses of chimeric constructs and alanine scan for DNA binding and other biochemical analyses in future, which should reveal the function of each part of VANC protein, including the repeated peptide motifs, in more solid manner.

Minor points:

1) Panels c, d, f, g of Figure 1 are not trivial to understand. The authors may consider using another way of illustrating the data.

Our response:

We added illustration for the loss of methylation in VANDAL family TEs in a more simple way (Supplementary Fig. 3). In addition, we showed raw methylation count data of all TEs longer than 1kb in Supplementary Data 1, so that readers can easily process the data by themselves. For Fig. 1, we agree that the formula is not easy to understand. In the legend, we added further explanation for the formula, as “This value shows the significance by weighing the change in the methylation ratios with root of the read number.” We would like to keep this way of presentation in Fig. 1, so that readers can easily compare the results with those in the previous publication (Fu et al 2013).

2) Numbered arrow heads on Figure 2b are not explained in the corresponding legend.

Our response:

The numbered arrowheads are explained in Figure 3. We added their explanation in legend of Figure 2b, too, so that readers are not puzzled.

3) Expressing VANC21 results in release of silencing at endogenous VANDAL21 elements. Does expression of VANC6 also lead to loss of silencing at VANDAL6 elements?

Our response:

We did RNA-seq for VANC6 TG. Other *VANDAL6* loci, as well as *VANDAL8* and *AT9TSD1* loci, are transcriptionally derepressed in trans. We added these results in the revised manuscript (Supplementary Fig. 4). We thank the reviewer for the suggestion.

4) In the discussion section, the authors state that rapid evolution of DNA repeats may have short-term advantage to escape from the host RNAi defense system. Although this is easy to conceive, how this would lead to “the long-term advantage to proliferate with minimum damage to the host” is less clear.

Our response:

We agree that the logic is not well explained. After “the long-term advantage to proliferate with minimum damage to the host”, we added “, because differentiation of anti-silencing systems would limit the number of proliferating TEs.” We believe the logic has become clearer.

Reviewer #2 (Remarks to the Author):

This reviewer is also very positive. We also thank the reviewer for constructive comments.

The manuscript submitted by Hosaka et al. describes the diversity and in some extent the evolutionary dynamics of "anti-silencing" TE-related sequences in Arabidopsis. VANC, one of the 3 ORF of the transposon has been shown to be involved in the silencing process by the same group in a previous publication. One open question concerns the specificity of VANC with Hi transposons, as well as the mechanisms of anti-silencing.

The authors first show that VANC protein is necessary for the demethylation of Hi transposons. They also show that the specificity is found in fast evolving regions of the TE sequence. These are tandem repeats, known to be highly dynamic.

I have no major objection for the publication of this manuscript in Nature communication. My only problem is that the manuscript lacks some estimates of the evolutionary rates :

The statement of "rapid evolution" is found in several instances throughout the manuscript and is not supported by precise figures. What is rapid ? for instance, the authors use the phylogeny of Vandal elements to show that the T-type motif is only found in *A. thaliana* and absent from *A. lyrata*. What is the divergent time between *A. thaliana* and *A. lyrata* ? Could the authors provide an estimate for the occurrence of T-motif ?

Our response:

We agree that quantitative discussion for the rapid evolution would be desirable. Separation between *A. thaliana* and *A. lyrata* has been estimated to be 5-10 million years ago (Koch & Matschinger 2007, Beilstein et al 2010) and average base substitution rate between these two species for neutral sites has been estimated to be in the order of 0.1 (Wright et al 2002). We added this information in Results section. This substitution rate cannot explain occurrence of target of motifs for VANC21 as well as VANC6 up to 20 copies within kilobases of regions around separation of these two species (Fig. 3D, Supplementary Fig. 11c). Dynamic evolution of tandem repeat should be the key for accumulation of multiple target motifs within the short regions. We added this discussion.

Similarly, the comparisons of tandem arrays among VANC paralogs in *A.t*

genomes show the frequent occurrence of VNTRs. The authors could estimate this frequency : 1 - by aligning paralogous sequence of complete elements 2 - compute divergence 3 - apply molecular clock to provide an estimation of VANDAL amplification in the genome, which would provide a time frame for diversification of tandem arrays.

Our response:

We agree that such estimation would be informative. It is very hard to align complete elements of different families of VANDAL, because their non-coding regions are very diverse. However, coding region of VANC and VANA are not so diverse. We therefore examined substitution rate of VANC and VANA paralogous genes (Supplementary Tables 3). It is possible however that speed of molecular clock changes in the proliferating TEs. Therefore, we also examined substitution rate of VANA and VANC between the same family of VANDAL copies between *A. lyrata* and *A. thaliana* and estimated the time since separation of specific families of VANDAL copies (Supplementary Tables 3, 4, Supplementary Fig. 14). By this procedure, we could roughly estimate that periods since separation of VANDAL families are in the order of 10 MY.

minor points :

- introduction : TE HTs are not rare in eukaryotes. See Peccoud et al., PNAS 2017 and ElBaidouri et al, Genome Research 2014.

Our response:

We added citation for these papers, with changing the expression to “Although horizontal transfer of TEs are known, that should be rare compared to that of viruses”.

- replace genera Arabidopsis by Arabidopsis genus.

Our response:

We corrected the term as suggested. We thank the reviewer for pointing that out.

Reviewers' comments:

Reviewer #1 (Remarks to the Author):

The authors satisfactorily addressed most of my previous comments, and added a nice set of data on VANC6 that further strengthens the author's claims regarding the specificity of the VANC/VANDAL system.

Although it is clear from the data that both VANC proteins and the DNA sequences recognized by these proteins evolve rapidly, I still believe that the data do not formally demonstrate coevolution between the DNA repeats and the tandemly arrayed peptide motifs of the VANC proteins. I fully agree that this is a likely hypothesis, but it should be mentioned as such in the manuscript.

Reviewer #2 (Remarks to the Author):

The authors have taken into account the remarks made in the first review. However, some points are not fully satisfactory. Not wanting to delay further the publication of this paper, I consider that it could not be released as long as the following points are not clarified :

This is about the evolutionary rate of the genes the authors are describing. I am still very uncomfortable with the use of "rapid" to describe this rate. The publication of wright et al. provides some estimates for Ka and Ks between thaliana and lyrata based on the analysis of 24 genes. These are under strong purifying selection as indicated by the Ka/Ks values. I am really not sure that using this data makes sense in the case of TE-related sequences. I think that some authors (devos et al., genome research 2002) have used a better estimate of Ks for TE-related sequences.

The true question is not whether the Ks is higher in the case of anti-silencing genes, but whether these genes are under selection ! Is it a case of co-evolution between TEs and the silencing genes, similarly to what is observed in Resistance/avirulence genes coevolution in plants for instance ? If this is the case, then I don't think that the systems described by the authors evolves particularly rapidly. Coevolution under strong selection may lead to a high evolutionary rate. Is this what the authors have in mind ? If yes, this should be phrased properly and the term rapid should be removed from the title.

Response to Reviewer #1

We are happy to see that the Reviewer #1 regards that the VANC6 data added during the revision have strengthened the manuscript.

We agree with the Reviewer #1 in that role of tandemly arrayed peptide repeats for the target specificities of VANC proteins is still not clear. To incorporate that point, we added the sentence “Investigation of the regions within the VANC proteins defining the target specificity will be a focus of next study.” in the Discussion section (p. 13, line 4-6 in the revised manuscript). In addition, we replaced the sentence “Taken together, our analysis of the VANC21 protein and the sequences recognized by this protein suggest that both of them evolve rapidly, and tandem repeat formation is a key to this rapid coevolution.” with “These tandem repeats within the VANC proteins might play roles for defining the target specificity, as is the case for the tandem repeats in their targets” (p. 11, line 5-7 in the revised manuscript), so that the suggestion for the role of the peptide motifs becomes more objective.

We thank the reviewer for helping us to improve the manuscript.

Response to Reviewer #2

As suggested by the Reviewer #2, we removed the word “rapid” from the title. We also modified the descriptions that the evolution of VANC proteins is rapid, because we agree with the reviewer that amino acid substitution rate of VANC proteins is not so high as those typically observed in genes under strong selection. We are speculating that tandemly arrayed peptides in the VANC proteins may contribute to the differentiation of target specificities of VANCs, but as mentioned in this and previous responses to the Reviewer #1, proving that experimentally will be a focus of future research.

In total, we removed majority (12 out of 22) of the uses of the term “rapid” or “rapidly”. Ten of them remained because we kept some of the descriptions that tandem repeat formation in the VANC target enables the rapid accumulation and synchronous evolution of multiple critical motifs. We also kept the description that centromeric tandem repeats can evolve rapidly and synchronously.

We thank the reviewer for helping us to improve the manuscript.